# Phenylpyrazolone-1,2,3-triazole Hybrids as Potent Antiviral Agents with Promising SARS-CoV-2 Main Protease Inhibition Potential

**DOI:** 10.3390/ph16030463

**Published:** 2023-03-20

**Authors:** Arafa Musa, Hamada S. Abulkhair, Ateyatallah Aljuhani, Nadjet Rezki, Mohamed A. Abdelgawad, Khaled Shalaby, Ahmed H. El-Ghorab, Mohamed R. Aouad

**Affiliations:** 1Department of Pharmacognosy, College of Pharmacy, Jouf University, Sakaka 72341, Saudi Arabia; 2Pharmaceutical Organic Chemistry Department, Faculty of Pharmacy, Al-Azhar University, Nasr City, Cairo 11884, Egypt; 3Pharmaceutical Chemistry Department, Faculty of Pharmacy, Horus University—Egypt, International Coastal Road, New Damietta 34518, Egypt; 4Chemistry Department, College of Sciences, Taibah University, Al-Madinah Al-Munawarah 41477, Saudi Arabia; 5Department of Pharmaceutical Chemistry, College of Pharmacy, Jouf University, Sakaka 72341, Saudi Arabia; 6Department of Pharmaceutics, College of Pharmacy, Jouf University, Sakaka 72341, Saudi Arabia; 7Department of Chemistry, College of Science, Jouf University, Sakaka 72341, Saudi Arabia

**Keywords:** 1,2,3-triazole, click chemistry, pyrazolone, SARS-CoV-2, antiviral activity, molecular modeling, molecular dynamics simulation

## Abstract

COVID-19 infection is now considered one of the leading causes of human death. As an attempt towards the discovery of novel medications for the COVID-19 pandemic, nineteen novel compounds containing 1,2,3-triazole side chains linked to phenylpyrazolone scaffold and terminal lipophilic aryl parts with prominent substituent functionalities were designed and synthesized via a click reaction based on our previous work. The novel compounds were assessed using an in vitro effect on the growth of SARS-CoV-2 virus-infested Vero cells with different compound concentrations: 1 and 10 μM. The data revealed that most of these derivatives showed potent cellular anti-COVID-19 activity and inhibited viral replication by more than 50% with no or weak cytotoxic effect on harboring cells. In addition, in vitro assay employing the SARS-CoV-2-Main protease inhibition assay was done to test the inhibitors’ ability to block the common primary protease of the SARS-CoV-2 virus as a mode of action. The obtained results show that the one non-linker analog **6h** and two amide-based linkers **6i** and **6q** were the most active compounds with IC_50_ values of 5.08, 3.16, and 7.55 μM, respectively, against the viral protease in comparison to data of the selective antiviral agent GC-376. Molecular modeling studies were done for compound placement within the binding pocket of protease which reveal conserved residues hydrogen bonding and non-hydrogen interactions of **6i** analog fragments: triazole scaffold, aryl part, and linker. Moreover, the stability of compounds and their interactions with the target pocket were also studied and analyzed by molecular dynamic simulations. The physicochemical and toxicity profiles were predicted, and the results show that compounds behave as an antiviral activity with low or no cellular or organ toxicity. All research results point to the potential usage of new chemotype potent derivatives as promising leads to be explored in vivo that might open the door to rational drug development of SARS-CoV-2 Main protease potent medicines.

## 1. Introduction

Single-stranded RNA Coronaviruses have been linked to mild to severe respiratory illnesses [1,2,3]. In 2002, human coronaviruses were recognized as the etiological agents for the global epidemic of the severe acute respiratory syndrome (SARS). At that time, SARS caused the death of more than 800 individuals worldwide. Shortly after, the Middle East Respiratory Syndrome (MERS-CoV) caused an additional wave of lethal respiratory diseases. More recently, the World Health Organization identified COVID-19 as a pandemic disease on 11 March 2020, and it promptly spread worldwide. More than 657,100,000 cases have been verified up to the present moment. This outbreak led to the death of over 6,657,000 individuals, representing a fatality rate of almost more than 1% until the 4 January 2023 [4]. Up till now, no approved effective drugs have been well-recognized to overcome COVID-19, making the discovery of novel powerful antiviral medication an urgent need.

Despite multiple attempts to develop antiviral drugs, especially in response to the rapidly developing pandemic COVID-19, viral diseases remain an increasing global public health concern [5]. Management of patients with COVID-19 still relies on the use of supportive, symptomatic, and repurposed drugs as an indispensable option [6]. Therefore, the exploration of anti-COVID-19 agents is essential to eradicate this pandemic. In the confrontation of this challenge, and over the last few years, numerous studies of ligand and structure-based computational methodologies were applied for the development of a virtual screening approach against libraries of synthetic and natural products that may help in the prediction of potential antiviral agents targeting COVID-19 [7]. Furthermore, several heterocyclic compounds have been developed that mainly target the COVID-19 Main protease [8]. Among these, nitrogen-containing heterocycles including triazoles and pyrazoles have been recognized as promising antivirals against coronaviruses, and their activity was reported to be mediated by targeting the coronavirus’s main protease (Main protease) [9,10,11,12]. This class of nitrogenous heterocycles plays a substantial role in the development of druggable candidates because of its ability to form hydrogen bonds with certain biological targets [13,14]. Moreover, 1,2,3-triazoles resist metabolic degradation relative to other heterocycles.

Two years ago, six 1,2,3-triazole derivatives (Figure 1), have been evaluated in silico against the Main protease of COVID-19. The results show that compound **1** presents a better interaction with the selected biological target than the co-crystallized ligand. Compound **1** with benzofuran and isatin moieties exhibits the highest energy and the best interaction pattern within the active site of the coronavirus main protease. These include four hydrogen-bonding interactions and four electrostatic interactions between the 1,2,3-triazole ring and the essential amino acid residues of the protein target site [9,15]. Moreover, studying the antiviral activity of fused 1,2,3-triazole derivatives **2** and **3** against human coronavirus reveals that the coronavirus’s main protease is the biological target of the antiviral activity of these derivatives [10,16]. One year earlier, the non-nucleoside inhibitory potential of triazole **4** was studied against the hepatitis C virus. Compound **4** demonstrates an EC_50_ value of 1.163 nM with low cytotoxicity in a subgenomic replicon experiment [17]. On the other hand, certain triazole-dihydropyrimidinone hybrids were synthesized by the application of a copper-catalyzed azide-alkyne cycloaddition reaction, and their antiviral activity was assessed against the human varicella-zoster virus. Without measurable cell growth inhibition, hybrid molecule **5** shows a promising antiviral potential, with an EC_50_ value of 8.38 μM against the varicella-zosterella zoster virus. Replacement of the triazole-attached phenyl rings in **5** with 4-NO_2_-phenyl (compound **6**) enhances the antiviral potency (EC_50_ 3.62 μM) and reduces the cytotoxicity [18]. 

On the other hand, pyrazolone is a pharmacophoric nitrogenous ring presented in several molecules with verified antiviral activity, particularly against coronaviruses [19,20,21,22,23,24,25]. It is interesting to note that pyrazolone derivatives **7** and **8** (Figure 2) displayed powerful inhibitory effects against the SARS-CoV Main protease with low micromolar IC_50_ values of 5.50 and 6.80 μM, respectively [20]. Moreover, pyrazolone analogs and **10** have been evaluated as inhibitors of the Main protease of SARS-CoV and MERS-CoV. Both compounds exert excellent inhibitory effects against SARS and MERS with IC_50_ values of 6.40 and 5.80 μM (**9**), 5.80 and 7.40 μM (**10**), respectively [22]. In our most recent investigations [23,26] a phenylpyrazolone scaffold exhibited good antiviral activity against SARS-CoV infection via a non-covalent inhibition of the main protease, the target protease. That investigation reported the formation of more rigid analogs through benzochromene formation with a terminal aromatic tail and two NH_2_ and CN functionalities (compound **11**). The behavior of these derivatives as antiviral agents was due to the good placement of compounds within the target pocket but showed some steric clashes. This inspired us to explore a chemistry change by combining a phenylpyrazolone scaffold with more flexible moieties that could compensate for the drawbacks in the previous study (Figure 2).

### Aim of the Work

Compounds incorporating 1,2,3-triazole, pyrazole, acetamide, and α, β-unsaturated carbonyl fragments (Figure 3) show effective covalent binding potential with key amino acid residues in the protein binding site [27,28,29]. These key amino acid residues include His41, Gly143, Cys145, Glu166, His163, His164, Gln189, and Thr190. 

One strategy for the design of a suggested drug candidate is to combine two pharmacophoric fragments of more than one well-known bioactive molecule belonging to a specific therapeutic class in one new scaffold [30,31,32,33]. For example, 1,2,3-triazole-based heterocycles were perfectly applied to develop numerous scaffolds with potent antiviral activity [16,17,18]. On the other hand, the penylpyrazolone scaffold represents a class of nitrogenous compounds with interesting antiviral activity [19,20,21,22]. Hybrids that contain or are linked by a 1,2,3-triazole ring in their structures are considered motivating molecular hybrids that offer a wide array of pharmacophore characteristics. Herein, given the aforementioned features and in continuation of our previous research in the design of suggested bioactive heterocycles [34,35,36,37], we have attempted to synthesize novel 1,2,3-triazoles linked to pyrazole moiety for further investigation as lead compounds in medicinal chemistry (Figure 3). Also, we report the in vitro antiviral evaluation of novel 1,2,3-triazole-pyrazolone hybrid molecules on COVID-19 [38]. It was reported that the complex of COVID-19 Main protease with the co-crystallized ligand, N3, consists of three chains, A, B, and C. Among these, A and C are the only two chains involved in the interaction. These interactions involve hydrogen bindings with the key amino acid residues Thr190, Glu166, Gln189, His164, His163, His41, and Gly143 [28,39]. In our work, the placement of pharmacophoric fragments that contribute to the stability of the scaffold similarly to the bound ligand, including phenoxy and 1,2,3-triazole, was identified within the binding pocket of the main protease target through interactions with the key residues Gln189 and Met49 by hydrogen bonds. In addition, the catalytic Cys145 contributes to the stabilization of the novel structures through stable hydrogen bonding with the NH part of the amide linker, which may help with the activity within the S1 subsite. As appeared from the structure of designed compounds, there is tolerance between all of the components of the target compounds (phenylpyrazolone, arylidene, active 1,2,3-triazole, amide linker, and terminal aromatic part) within the active site and the reflected inhibitory activity against both viral propagation and the Main protease target. 

## 2. Results and Discussion

### 2.1. Chemistry

As a continuation of our interest in the click chemistry and biological properties of 1,2,3-triazoles [40,41,42,43,44,45], we have attempted to design and synthesize novel 1,2,3-triazoles linked to pyrazolone moiety for further investigation as lead compounds in medicinal chemistry. Thus, the general synthetic protocols adopted for the synthesis of the targeted 1,2,3-triazole-pyrazolone hybrids are represented in Figure 1, Figure 2 and Figure 3. Hybrid molecules that contain or are linked by a 1,2,3-triazole building block are usually referred to as “lead compounds”; these molecular hybrids offer a wide array of pharmacophore characteristics. Given the aforementioned features and researchers’ recent interest in the chemistry of 1,2,3-triazole molecules [13,46,47,48], novel 1,2,3-triazoles linked to pyrazole moiety for further investigation were designed and synthesized as lead compounds in medicinal chemistry as a continuation of our interest on these tunable scaffolds and their biological properties [49,50,51,52,53].

Thus, the protocols of the synthesis of the targeted hybrids are illustrated in Figure 1, Figure 2 and Figure 3. The precursor alkyne derivatives **4a**–**b** used in this study were obtained through the condensation of the commercially available pyrazole-3-one derivatives **3a**–**b** with the synthesized propargylated benzaldehyde **2** using piperidine as a basic reagent (Figure 1). The propargylated benzaldehyde de **2** was in turn obtained through base-catalyzed (potassium carbonate) alkylation of 4-hydroxybenzaldehyde (**1**) with propargyl bromide according to the literature [54].

Based on the spectroscopic results, the structures of the pyrazolone-based alkyne side chains **4a**–**b** were revealed. Their IR spectra clearly show the absence of the absorption bands belonging to the C=O and C–H of the aldehyde group, and the presence of characteristic bands at 2140–2150 and 3280–3290 cm^−1^ confirming the presence of the C≡C and ≡C–H groups in their structures. In addition, no signal was recorded in their ^1^H NMR for the aldehydic proton which confirms the condensation step. The spectra also revealed distinct singlets at δ_H_ 8.97–10.82 ppm and 3.66–3.69 ppm, attributed to new sp2-CH (C=C–H) linker and Sp(≡C–H) protons, respectively. Moreover, the investigation of ^13^C NMR spectra supported the resulting structures through the recording of signals around δ_C_ 79.33–79.39 ppm attributed to the acetylenic carbons (C≡C) and absence of the carbonyl aldehyde carbons. In the Appendix A, all additional protons and carbons are recorded and discussed. 

The aromatic azides **5a**–**g** involved in this study were prepared by diazotization of different substituted aromatic anilines in situ catalyzed by a mixture of sodium nitrite in HCl and afterward treated with sodium azide according to the reported procedure [55]. On the other hand, the functionalized aniline/benzylacetamide azides **5h**–**o** have been successfully prepared through the reaction of an appropriate amine reagent (substituted anilines and benzylamine) with bromoacetyl bromide in basic triethylamine medium using DCM as solvent followed by the addition of sodium azide in a mixture of acetone: water (4:1, *v*/*v*) as reported in the literature [56].

**Figure 3 pharmaceuticals-16-00463-f003:**
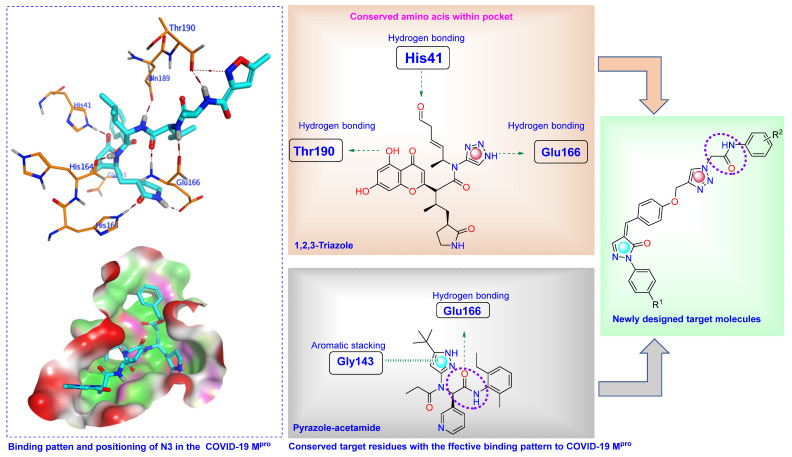
Strategy design of target molecules.

**Scheme 1 pharmaceuticals-16-00463-sch001:**
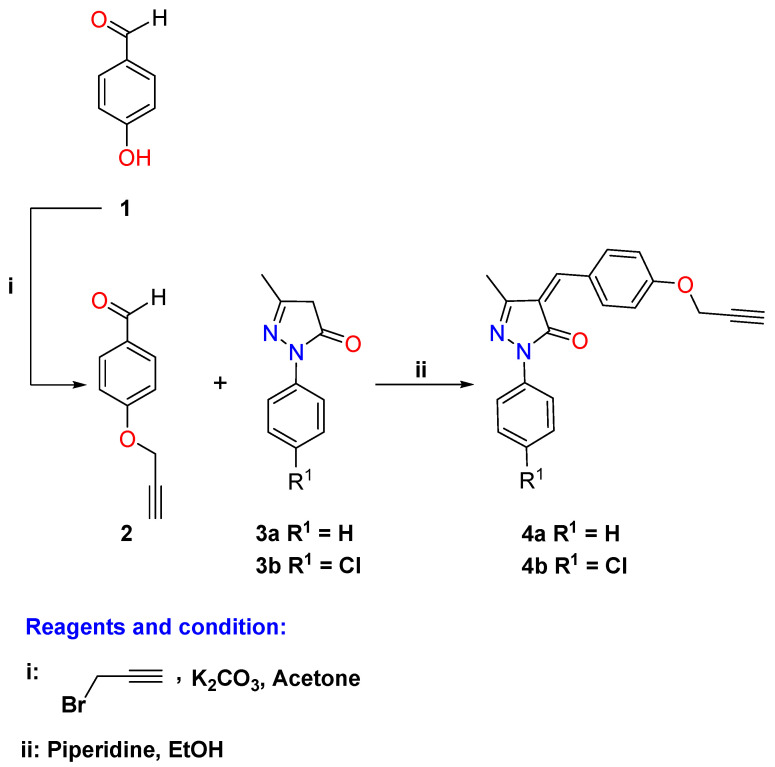
Synthesis of the propargylated pyrazolone derivatives **4a**–**b**.

**Scheme 2 pharmaceuticals-16-00463-sch002:**
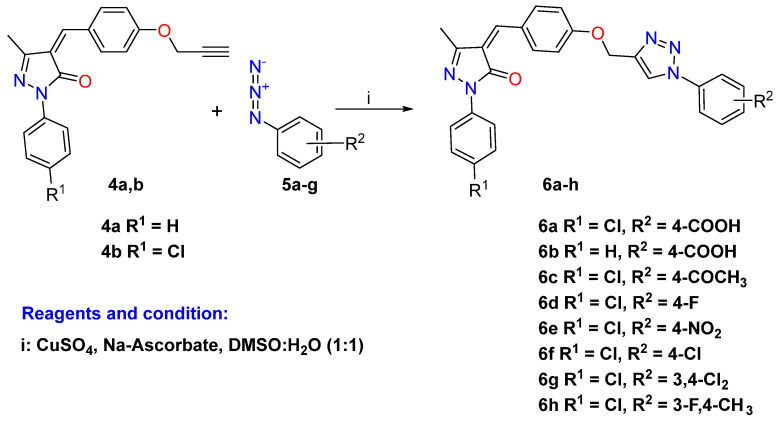
Click synthesis of 1,2,3-triazole hybrids carrying aromatic units **6a**–**h**.

**Scheme 3 pharmaceuticals-16-00463-sch003:**
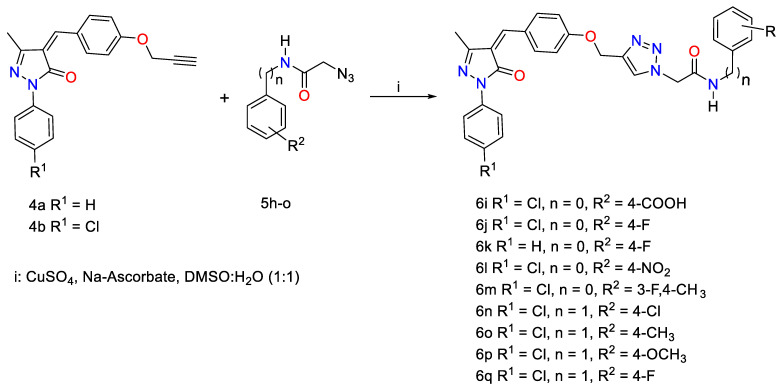
Click synthesis of pyrrazole-1,2,3-triazole hybrids carrying phenyl acetamide side chain **6j**–**q**.

Click chemistry introduced green synthetic protocols with higher yields for the synthesis of 1,2,3-triazole through the discovery processes in the medicinal chemistry discipline via azide and alkyne cycloaddition reactions. Thus, the CuAAC 1,3-dipolar cycloaddition reaction of the alkyne-pyrazole derivatives **4a**–**b** with focused aromatic azides **5a**–**g** was carried out in the presence of CuSO_4_ and Na_2_CO_3_ in a solvent mixture of DMSO/H_2_O at room temperature and afforded the desired pyrazolone-1,2,3-triazole hybrids carrying substituted aromatic units **6a**–**h** in moderate to good yield (86–92%), as described in Figure 2. 

Similarly, novel pyrazolone-1,2,3-triazole hybrids **6j**–**q** carrying phenyl acetamide side chains were designed as a second set and synthesized through the click ligation of the pyrazolone-alkyne building blocks **4a**–**b** with some functionalized phenyl/benzyl acetamide azides **5h**–**o** under optimized Cu(I)-assisted 1,3-dipolar cycloaddition protocol (Figure 3). 

The success of the click reactions was supported by the investigation of the spectroscopic data of the results in the 1,2,3-triazole-pyrazolone hybrids **6a**–**h** and **6i**–**q,** including IR, ^1^H NMR, ^13^C NMR, and elemental analyses. As a result, no signals were observed in the alkyne area in their IR spectra, providing strong proof for the cycloaddition reaction’s success. Additionally, their ^1^H NMR spectra disclosed the absence of the acetylenic protons (≡C–H) and the presence of diagnostic singlet around δ_H_ 8.27 to 9.15 ppm attributed to the CH-1,2,3-triazolyl protons, validating the cycloaddition reactions. 

The spectra also revealed distinguishable singlets and/or triplets ranging between δ_H_ 4.29 to 5.46 ppm related to the protons of one, two, and/or three methylene (–CH_2_–) bridges constructing the adducts 6j–q. The presence of the acetamide group in the structure of this set of click products 6j–q was also supported by the resonance of characteristic NH-protons recorded at δ_H_ 8.79 to 10.82 ppm. Moreover, the click reactions were also validated by the investigation of their ^13^C NMR spectra. Thus, the absence of signals attributed to the alkyne chain (C≡C) and the presence of extra signals belonging to aromatic carbons of the phenyl rings are good pieces of evidence for the success of the cycloaddition reaction. In addition, the spectra of the click adduct 6i–q revealed new signals at δ_C_ 160.4 to 165.9 ppm assigned to the acetamide carbonyl carbons (CONH), supporting the proposed structures. All remaining signals are listed and detailed in the experimental section.

### 2.2. Pharmacological Testing

#### 2.2.1. Inhibitory Assay of SARS-CoV-2 Main Protease

Because of its critical involvement in the polypeptide of virus catalysis, Main protease (or 3CLpro) is considered a feasible therapeutic target in SARS-CoV-2 [57,58]. Several Main protease inhibitors, including GC-376, PX-12, and carmofur, were identified as covalent inhibitors that alter Cys145’s catalytic function with powerful enzymatic inhibitory effect and cellular antiviral activity, with reactive nitrogen heterocyclic as pyrrolidone, imidazole, or pyrimidine that mimics the glutamine P1 position [59,60]. Furthermore, many non-covalent inhibitors such as ML188 were discovered as SARS-CoV-2 Main protease inhibitors [61]. Moreover, several studies have been conducted on the ML188 series to optimize the enzymatic potency against the SARS-CoV-2 main protease until recently finding the novel non-covalent Main protease inhibitor (23R) with potent cellular antiviral activity [61]. Pyrazolone-triazole hybrids were investigated in vitro for suppression of SARS-CoV-2 Main protease activity and compared to the well-known antiviral GC-376 as a selective Main protease inhibitor. The structure of analogs is distinct according to the terminal part attached to the triazole ring; one ends with lipophilic aryl substituted groups (**6a**–**h**), and the other ends with an amide linker and terminal-substituted aryl functionality (**6i**–**q**). Accordingly, the results exhibit a better activity profile for the second class with amide linker than the other. As shown in Table 1**,** it appears that the chlorophenylpyrazolone analogs show a very promising potency, which is better than the unsubstituted phenylpyrazolone. These findings confirm the antiviral activity of such a Cl-phenylpyrazolone scaffold better than plain phenylpyrazolone itself as reported in our previous study [23]. As seen in Table 1, three main classes appear with distinct performances against the studied viral protein. The first one includes the potent analogs **6i**, **6h**, and **6q** that show better inhibition activity on the target enzyme than the reference drug GC-376. The second class, comprising **6d**, **6e**, **6j**, **6m**, **6o**, and **6p**, has relatively lower activity than that of the main protease selective inhibitor with IC_50_ values range of 25.70–49.39 μM. The third class comprises the remaining phenylpyrazolone-1,2,3-triazole hybrids with much weaker activity compared with the control drug. Among all the new phenylpyrazolone-1,2,3-triazole hybrids, the 4-chlorophenylpyrazolone derivative **6h** is the most potent one with ether linkage. Likewise, the 4-chlorophenylpyrazolone derivative with carboxylic acid functionality **6i** is the most potent member among all the new amide-linked triazoles. Fluorine substituents on triazole-linked phenyl exert better impact on the activity compared with chlorine and nitro substituents. This can be clearly recognized upon comparing the superior activity of analogs **6h**, **6q**, and **6m** with IC_50_ values of 5.08, 7.55, and 25.7 μM, respectively, with those of **6a**–**6d** (IC_50_ values ≥ 76.91 μM). The relatively high micromolar-inhibitory concentration of CG-376 obtained in the current study (12.85 μM) relative to those reported in earlier works [62,63] might be attributed to lab-based handling, sensitivity, and the type of kit used. 

#### 2.2.2. SAR Analysis of Target Compounds

According to the antiviral results, the target compounds possess powerful antiviral efficiency against SARS-CoV-2 Main protease. The priority structure-activity relationship results were extracted based on the anti-SARS-CoV-2 Main protease activity as indicated in (Table 1). On one hand, the attachment of a *para*-chlorophenyl fragment to the pyrazolone ring significantly increases main protease the activity. This is obvious in the comparison between the SARS-CoV-2 Main protease inhibitory activity under the effect of **6b** and **6k** (with unsubstituted aryl) with those of all other derivatives. On the other hand, the attachment of a di-substituted aryl fragment into the triazole ring markedly enhances the activity. For instance, the di-substituted derivative **6m** shows a better SARS-CoV-2 main protease inhibitory effect. Moreover, the di-substituted derivative **6h** demonstrates better anti-SARS-CoV-2 main protease activity than the control inhibitor. Lastly, the insertion of an acetamide linker has revealed a positive impact on the overall SARS-CoV-2 Main protease inhibitory activity. Accordingly, three *N*-aryl 1,2,3-triazoles (**6a**, **6c**, and **6n**) show comparable inhibitory activity. They inhibit SARS-CoV-2 Main protease with IC_50_ values of 76.91, 81.14, and 86.61 μM compared with 82.17 lopinavir. Even more, six derivatives (**6d**, **6e**, **6j**, **6m**, **6o**, and **6p**) exhibit better SARS-CoV-2 main protease inhibitory activity. They present IC_50_ values of 42.45, 30.08, 49.39, 25.7, 43.31, and 23.73 μM, respectively. More interestingly, three 1,2,3-triazole-pyrazolone hybrid molecules (**6h**, **6i**, and **6q**) inhibit the SARS-CoV-2 Main protease activity more strongly than the specific antiviral agent GC-376 with IC_50_ values of 5.08, 3.169, and 7.55 μM compared with 12.85 μM. Graphical representations of the SAR analysis are given in Figure 4 and Figure 5.

### 2.3. Antiviral Effect against SARS-CoV-2 Activity

Investigation of the newly synthesized compounds for their antiviral capacity to reduce SARS-CoV-2 viral duplication, a cellular antiviral assay for hybrid compounds against SARS-CoV-2 at two doses (1 and 10 μM) on infected Vero E6 cells was done utilizing traditional cell culture with a real-time PCR. The MTT test was used to assess cell viability following analog or remdesivir treatment in Vero E6 cells (CCL-8; ATCC origin donated by the Virology Sector, VACSERA, Giza, Egypt; https://www.vacsera.com (accessed on 21 April 2022)) infected with SARS-CoV-2 at a multiplicity of infection (MOI) of 0.01. At a dose of 60 M, remdesivir was employed as a positive control; it demonstrated no cytotoxicity to Vero E6 cells. Except for five compounds (**6a**, **6i**, **6n**, and **6p**), all were well-tolerated and have IC_50_ values of greater than 30 M for the Vero E6 cell lines examined. However, 4-COOH phenyl analog (**6i**) inhibits Main protease activity well; it is less lethal to Vero E6 cells, with an IC50 value of 22.1 μM and promising inhibition for viral growth at 10 μM by 64% compared to the reference drug. The most potent analogs 4-F and 4-COOH; **6i** and **6q** show unapparent cytotoxicity with IC_50_ values above 20 μM for Vero E6 cell lines and better viral growth inhibition by 79 and 64%. They also have the same enzymatic inhibition and cellular antiviral efficacy as the most effective SARS-CoV-2 Main protease inhibitors. Remdesivir and GC-376 were applied as positive controls, and the findings reveal that they effectively suppress viral replication at concentrations of 10 and 1 μM, respectively. Interestingly, **6q**, **6h**, and **6i** were all effective inhibitors of Main protease recombinant SARS-CoV-2 Main protease (IC_50_: 3.16, 5.08, and 7.55 μM, respectively) and outperform the others in the live viral replication experiment, reaching 79%, 64.2%, and 63 at 10 μM, respectively. In general, none of the pyranopyrazole compounds inhibit viral replication to 50% at the lowest dose (1 μM) (Table 2, Figure 6). Furthermore, several pyranopyrazole compounds do not perform well in the cellular antiviral assay, and their relative ability to suppress viral replication does not always correspond directly with in vitro inhibitory characteristics against main protease [64]. Compound **6a** inhibits cellular antiviral activity the least (19.27% at 10 μM), which is consistent with its low Main protease enzymatic inhibition. (IC_50_ values of 76 μM). Compounds **6p** and **6m** prove their moderate target Main protease inhibitory effect rather than their viral replication effect with ≈50% at 10 μM. Generally, low matching of SAR data of antiviral effects of target compounds with the above main protease inhibitory data was observed, which might be attributed to the different modes of action of some derivatives, which needs some more extensive biological assays.

### 2.4. Computational Studies

#### 2.4.1. Molecular Docking Analysis

The active site is situated between the I (amino acids 10–99) and II (amino acids 100–182) barrel domains. Protein dimerization and the construction of a bundle of alpha helices are carried out by the other third domain, which is made up of residues 198 to 306 [55]. The catalytic dyad and dimerization, which complete the active site by bringing Ser161 of the second dimer protomer into proximity with Glu166 and facilitating the development of the substrate specificity pocket and the oxyanion hole, are formed by two major conserved residues, His41 and Cys145 [56]. Two fragments, phenoxy and triazole, were identified which contribute to the stability of the scaffold within the binding pocket through interactions with the key residues Gln189 and Met49 by hydrogen bonds. In addition, the catalytic Cys145 contributes to the stabilization of the compound through stable hydrogen bonding with the NH part of the amide linker which might help in the activity within S1′ subsite. According to reports, the S2 subsite exhibits higher flexibility than the other subsites, accepting smaller substituents in peptide-based inhibitors but favoring leucine or other hydrophobic residues [64]. A common pattern of an aromatic ring formation, stacking interactions with Met49, which drives the terminal phenylpyrazolone component to near to S1’ site, causes many fragments bonded at this point to be referred to as the “aromatic wheel”. The compound fragment phenoxy had an expansion to the residue Gln189 in S3 subsite through aromatic stacking interaction. The interaction analysis of the most prominent active compound **6i** revealed non-covalent interactions and the contribution of most chemical moieties in the activity tolerate the pocket with all catalytic sites, as shown in Figure 7.

#### 2.4.2. Molecular Dynamic Analysis

Herein, we looked at the protein’s secondary structure of main protease coupled to the **6i** new active analog before and after docking to understand the conformational changes and their stability. The main protease secondary structure has 49,540 total atoms, a zero-net charge, 51.040 mM of Na+ ions, 53.956 mM of Cl ions, and 12,468 water molecules when it was put through an MD simulation test.

100 ns MD runs were used to estimate the RMSD and RMSF for the complicated reference and **6i** analog forms. The ligand’s binding can prevent the protein from unfolding and stabilize it [65]. To better comprehend the conformational changes brought on by ligand contact, the RMSD record of the main protease response before and after docking of the **6i** triazole derivative, which found equilibrium within 20 ns, is shown in Figure 8 (up for the reference drug, down for **6i**). In the MD simulation, the target protein was confirmed to be stable at RMSD 20, which is a respectable value for protein structures that demonstrate the complex stability.

The N- and C-terminal residues oscillates above 2.0 for the reference drug and below 2.0 for the **6i** derivative, according to the RMSF plot for main protease using the reference and **6i** compounds. The predicted RMSF of the protein-ligand complex (Figure 8A) is less than 2.0, indicating conformational stability during the simulation. The other secondary structures remain consistent throughout the trajectory. In contrast to the complex with **6i**, which has 25.96% helices and 9.77% strands, the Main protease complex with reference (Figure 8B) includes 25.27% helices and 8.81% strands. The little unfolding of the α-helices throughout the complex’s MD is shown by the minor rise in the percentage of β-strands.

The intermolecular interactions of the Main protease complexes were examined during the MD simulation, and the results are shown in Figure 9A–C. This analysis confirms the molecular docking findings by demonstrating that the studied trajectories contain many potential contacts of both polar and non-polar interactions. Furthermore, Cys106 os the conserved residue among the multiple interactions presented, as seen by the plot of interaction percentages vs. binding site residues (Figure 9B,C) (interaction fraction greater than 1.0). The plot, shown in Figure 10, demonstrates that the binding site residues of the Main protease complex, Cys106, Asp167, Asp109, Lys48, and Glu107, were involved in hydrogen bonding at various simulation times, supporting the conclusion that the docking results are compatible with the MD results. The **6i** compound’s hydrophobic interactions with the main protease complex are mediated by the triazole, linker, and terminal pyrazolone fragments.

#### 2.4.3. Prediction of Drug-Likeness and ADME Properties

The adverse effects risk, such as reproductive consequences, irritation, tumorigenicity, and mutagenicity alongside the drug-relevant characteristics, such as cLogP, LogS (solubility), MW, drug-likeness, and overall drug-score, were estimated using OSIRIS Property Explorer. The two potent and low active analogs **6i** and **6f** were compared to the published drug in in silico physicochemical and toxicological assessments using the OSIRIS Property Explorer tool. cLogP, LogS (solubility), MW, drug-likeness, and total drug-score parameters are scored and color-coded, as shown in Table 3. Properties with a high risk of unfavorable consequences, such as mutagenicity, tumorigenicity, and impacts on reproductive physiology), are also rated and color-coded. Red indicates a high risk of toxicity, and green indicates no risk, hence the data on toxicity risks reflect behavior that is consistent with the substance. It is interesting to note that the potential drug-likeness values of compound **6i** are much greater than those of compound **6f** and GC-376, which has a negative value of 28.05. Nevertheless, the in silico analysis performed by OSIRIS Property Explorer revealed that adding 4-COOH to the aryl ring can maintain a low risk of tumorigenic and mutagenic toxicity while enhancing. Generally, the values of drug-score of the representative compound **6i** (0.24) are less than GC-376 (0.37).

## 3. Concluding Remarks

A series of phenylpyrazolone with 1,2,3-triazole and lipophilic aromatic terminals were designed and tested for dual antiviral and Main protease inhibitory action in this work. The majority of the synthesized compounds are equipotent or have greater activity than the reference inhibitors. Most of the compounds exhibit both viral growth inhibition and higher anti-Main protease activity compared with positive control drugs. In addition, amide linker-based compounds reveal more antiviral effects than non-amide ones. Compound **6i** has the most potent antiviral effect (64% inhibition) and Main protease inhibitor with IC_50_ values of 3.1 μM. Based on our results, compound **6i** is indicated to have considerable potential as a new pyrazolone-1,2,3-triazole hybrid as the lead compound for discovery of novel antiviral drugs for management of COVID-19 infection. The **6i** acid derivative that demonstrates the greatest drug-likeness and drug-score values are considered as a lead for the creation of antiviral medicines in future.

## 4. Materials and Methods

### 4.1. Chemistry

None of the reagents or solvents used were further refined and were the best possible quality of analytical reagents. The melting points are uncorrected and were established using Stuart Scientific SMP1, Bibby Scientific, Stone, Staffordshire UK. Spots were seen using a UV light during TLC on UV fluorescent Silica gel Merck 60 F254 plates (254 nm, Sigma-Aldrich Chemie, Taufkirchen, Germany). With a 400–4000 cm^−1^ range, the SHIMADZU FTIR-Affinity-1S spectrometer (PerkinElmer, Seer Green, Beaconsfield, UK) was employed for identification of the major functional groups. Tetramethyl Silane (TMS) was used as an internal reference in a Bruker spectrometer (400 MHz, PerkinElmer, Seer Green, Beaconsfield, UK) to capture the NMR spectra. A GmbH-Vario EL III Elementar Analyzer (ELTRA GmbH, Haan, Germany) was applied for elemental analyses.

#### 4.1.1. Synthesis and Characterization of 4-(prop-2-yn-1-yloxy)-Benzaldehyde (2)

A mixture of 4-hydroxybenzaldehyde (**1**) (10 mmol), potassium carbonate (12 mmol) and propargyl bromide (12 mmol) in acetone (25 mL) was heated under reflux for 2 h till the consumption of the start. TLC was used for monitoring (hexane-ethyl acetate). Cooling of the mixture was done by crushed ice water. The resultant precipitate was filtered, water-washed, dried, and crystallized from ethanol to get the desired *O*-propargylated benzaldehyde **2** as yellowish crystals in 94% yield, mp: 80–81 °C. IR (*υ*, cm^−1^): 1550 (C=C), 1710 (C=O), 2150 (C≡C), 2850 (CH-Al), 3080 (CH-Ar), 3240 (≡CH). ^1^H NMR (400 MHz, CDCl_3_) δ_H_ = 2.59 (t, 1H, *J* = 4.0 Hz, ≡CH), 4.79 (d, 2H, *J* = 4.0 Hz, OCH_2_), 7.08 (d, 2H, *J* = 8.0 Hz, Ar-H), 7.87 (d, 2H, *J* = 8.0 Hz, Ar-H), 9.90 (s, 1H, CHO). ^13^C NMR (100 MHz, CDCl_3_): δ_C_ = 55.94 (OCH_2_); 76.44 (C≡CH); 77.54 (C≡CH); 115.16, 130.53, 131.94, 162.36 (Ar-C); 190.88 (C=O). Calculated for C_10_H_8_O_2_: C, 74.99; H, 5.03. Found: C, 75.23; H, 5.25.

##### (Z)-1-(4-Chlorophenyl)-3-Methyl-4-(4-(prop-2-yn-1-yloxy)Benzylidene)-1H-Pyrazol-5(4H)-One (4a)

It was obtained as orange pellets in 88% yield, mp: 134–135 °C. IR (*υ*, cm^−1^): 1570 (C=C), 1700 (C=O), 2150 (C≡C), 2930 (CH-Al), 3080 (CH-Ar), 3290 (≡CH). ^1^H NMR (400 MHz, DMSO-*d_6_*) δ_H_ = 2.31 (s, 3H, CH_3_), 3.69 (s, 1H, ≡CH), 4.96 (s, 2H, OCH_2_), 7.17 (d, 2H, *J* = 8.0 Hz, Ar-H), 7.48 (d, 2H, *J* = 12.0 Hz, Ar-H), 7.75 (s, 1H, H–C=C), 7.96 (d, 2H, *J* = 12.0 Hz, Ar-H), 8.68 (d, 2H, *J* = 8.0 Hz, Ar-H). ^13^C NMR (100 MHz, DMSO-*d_6_*): δ_C_ = 13.60 (CH_3_), 56.34 (OCH_2_); 79.37 (C≡CH); 79.43 (C≡CH); 115.54, 120.07, 124.31, 127.09, 128.61, 129.21, 137.13, 137.59, 148.79, 152.72, 161.94, 162.25 (Ar-C, C=O).

Calculated for C_20_H_15_ClN_2_O_2_: C, 68.48; H, 4.31; N, 7.99. Found: C, 68.65; H, 4.55; N, 8.09.

##### (Z)-3-methyl-1-Phenyl-4-(4-(prop-2-yn-1-yloxy)Benzylidene)-1H-Pyrazol-5(4H)-One (4b)

It was obtained as orange pellets in 90% yield, mp: 158–160 °C. IR (*υ*, cm^−1^): 1580 (C=C), 1690 (C=O), 2140 (C≡C), 2880 (CH-Al), 3060 (CH-Ar), 3280 (≡CH). ^1^H NMR (400 MHz, DMSO-*d_6_*) δ_H_ = 2.33 (s, 3H, CH_3_), 3.66 (s, 1H, ≡CH), 4.94 (s, 2H, OCH_2_), 7.15 (d, 2H, *J* = 8.0 Hz, Ar-H), 7.38–7.45 (m, 3H, Ar-H), 7.77 (s, 1H, H–C=C), 7.94 (d, 2H, *J* = 12.0 Hz, Ar-H), 8.67 (d, 2H, *J* = 8.0 Hz, Ar-H). ^13^C NMR (100 MHz, DMSO-*d_6_*): δ_C_ = 13.67 (CH_3_), 56.38 (OCH_2_); 79.33 (C≡CH); 79.41 (C≡CH); 114.79, 115.50, 119.82, 124.28, 127.02, 128.66, 129.24, 137.04, 137.48, 148.67, 152.66, 161.87, 162.18 (Ar-C, C=O). Calculated for C_20_H_16_N_2_O_2_: C, 75.93; H, 5.10; N, 8.86. Found: C, 75.78; H, 5.31; N, 8.99.

#### 4.1.2. General Click Procedure for the Synthesis of 1,2,3-Triazole-Pyrazolehybrids 6a–q

A solution of propargylated pyrazoles **4a**–**b** (1 mmol) in DMSO was added with stirring to copper sulphate solution (0.10 g) and sodium ascorbate (0.15 g) in water (10 mL) (10 mL). The suitable azide **5a**–**o** (1 mmol) was then added to the reaction mixture, which was agitated at room temperature for 24–40 h. The reaction was monitored by TLC (hexane-ethyl acetate). When the reaction was finished, the liquid was put into chilled water. The resulting precipitate was collected through filtering, washed with ammonium chloride solution, and recrystallized from ethanol/DMF to get the anticipated 1,2,3-triazoles **6a**–**q**. All molecular characterizations are described in the Appendix A.

### 4.2. Main Protease Inhibition Assay

The COV2-SARS-CoV-2 protease enzyme assay [66,67,68] was described in the manufacturing protocol (BPS Bioscience). The methodology is described in the Appendix A.

### 4.3. SARS-CoV-2 Antiviral Assay

The effect of target chemical treatments on SARS-CoV-2 viral load (SARS-CoV-2 isolate EGY/WAT-2 VACCERA) was assessed by the Real-Time PCR test to detect SARS-CoV-2 viral RNA [69,70]. Total RNA was extracted according to the instructions using the genesig^®^ Coronavirus SARS-CoV-2 Real-Time PCR Assay kit (Primer design^TM^ Ltd., Southampton, UK). All procedures are depicted in the Appendix A.

### 4.4. Molecular Docking Study

The simulation was conducted by utilizing the PDB crystal structure of SARS-CoV-19 Main protease for the target molecule **6i** in comparison to the reference bound ligand. (PDB: 5R80; https://www.rcsb.org/structure/5R80 (accessed on 21 April 2022)). Autodock 4.0.1.34 tool its graphical interface was utilized for the simulation [71] and (MOE 2014, molecular operating environment software) [72]. The docking protocol was detailed previously [73,74,75]. The pdb target was prepared by applying the default energy minimization protocol after removing the bound ligand and water molecules except the ones needed within the binding pocket. The reproducibility of the docking simulation was validated by docking of the bound ligand as control and the adjustment of method was achieved by calculation of RMSD and analysis of the binding data. Afterward, the **6i** compound was docked and output data was analyzed and represented [76].

### 4.5. Molecular Dynamics Simulation

MD simulations were performed by using templates from the 3D structural crystal of the SARS-CoV 19 main protease (PDB ID: 5R80) combined with the active compound **6i** vs. the reference bound drug. Desmond version 3.8 software and the OPLS2005 forcefield were used to execute the simulation [77]; see more details in Appendix A. The production molecular dynamics phase was performed for three independent 100 ns simulations in an isothermal-isobaric (NpT) ensemble at 300 K and 1 bar using Langevin dynamics.

## Data Availability

Data are contained within the article and Appendix A.

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
