# Peer review of "Phenylpyrazolone-1,2,3-triazole Hybrids as Potent Antiviral Agents with Promising SARS-CoV-2 Main Protease Inhibition Potential"

_pharmaceuticals, 2023, doi:10.3390/ph16030463_

Round 1

Reviewer 1 Report

The title of the manuscript does not adequately summarize the work described. The work disclosed is not a fragment-based drug design approach it is based on the superposition and merger/morphing of 2 previously described SARS-COV-2 Mpro inhibitors.

The manuscipt describes a set of 17 new compounds for which limited SAR mostly on the triazole substituent has been generated. These compounds have been designed based on the overlay of structures of 2 previously described SARS-COV-2 Mpro inhibitors, this is not a fragment based drug discovery approach. In the paper also the term `lead compound` is used incorrectly (e.g. page 3 last sentence, 3rd sentence on page 4).

The inhibitors described are not highly potent as claimed, the best derivatives display single digit micromolar activity which is a good but by far not an outstanding result. In addition the IC50 value for the reference compound CG376 as given in Table 1 is generally about > 10-fold off than what is described in the literature (e.g. Nat Commun 11, 4417, 2020), generally CG376 is described as a nanomolar  SARS-COV-2 Mpro inhibitor. Lopinavir is an  incorrect reference compound as it is a HIV-protease inhibitor and was optimized as such it is not a SARS-COV-2 Mpro inhibitor. Lopinavir in combination with ritonvavir has been used with controversial results to treat COVID 19. The author should also use the same reference compounds for the biochemical (Table 1) and the cellular assay (Table 2). Remdesivir was used as the reference compound for the cellular assay it would be appropriate to use GC376. Also the cellular data does not exhibit the same SAR as the biochemical assay, this should be commented and explained.

Generally the work described is rather suited for a short communication than for a full article. The most potent, single digit micromolar SARS-COV-2 Mpro inhibitors disclosed might offer an interesting starting point for further optimization but not more.  

Author Response

Referee 1

Journal: Pharmaceuticals (ISSN 1424-8247)

Manuscript ID: pharmaceuticals-2216659

Title: Phenylpyrazolone-1,2,3-triazole hybrids: A fragment-based drug design approach towards the discovery of potent antiviral agents with promising anti-SARS-CoV-2-MPro inhibition potential

Author's Reply to the Review Report (Reviewer 1)

(x) I would not like to sign my review report

( ) I would like to sign my review report

English language and style

( ) English very difficult to understand/incomprehensible

( ) Extensive editing of English language and style required

(x) Moderate English changes required

( ) English language and style are fine/minor spell check required

( ) I don't feel qualified to judge about the English language and style

Yes      Can be improved       Must be improved     Not applicable

Does the introduction provide sufficient background and include all relevant references?

( )        ( )        (x)       ( )

Are all the cited references relevant to the research?

(x)       ( )        ( )        ( )

Is the research design appropriate?

( )        (x)       ( )        ( )

Are the methods adequately described?

(x)       ( )        ( )        ( )

Are the results clearly presented?

( )        ( )        (x)       ( )

Are the conclusions supported by the results?

( )        ( )        (x)       ( )

Comments and Suggestions for Authors

The title of the manuscript does not adequately summarize the work described. The work disclosed is not a fragment-based drug design approach it is based on the superposition and merger/morphing of 2 previously described SARS-COV-2 Mpro inhibitors. The manuscipt describes a set of 17 new compounds for which limited SAR mostly on the triazole substituent has been generated.

  1. These compounds have been designed based on the overlay of structures of 2 previously described SARS-COV-2 Mpro inhibitors, this is not a fragment based drug discovery approach.

We meant here by a fragment-based approach the molecular hybridization protocol of different azoles; pyrazole-linker-triazole with terminal aromatic fragments.

  1. In the paper also the term `lead compound` is used incorrectly (e.g. page 3 last sentence, 3rd sentence on page 4).

Thanks for your comment, they are identified as precursors containing pharmacophores used in our study. We corrected them in MS.

  1. The inhibitors described are not highly potent as claimed, the best derivatives display single digit micromolar activity which is a good but by far not an outstanding result.

All regards for you, our study introduced a novel scaffold with multi-azole ring of antiviral activity, however, the activity is not outstanding, but this is the first paper published this scaffold and proven their anti-COVID activity. In the next work, we have different protocols for optimization of these active derivatives to potent ones.

  1. In addition the IC50 value for the reference compound CG376 as given in Table 1 is generally about > 10-fold off than what is described in the literature (e.g. Nat Commun 11, 4417, 2020), generally CG376 is described as a nanomolar  SARS-COV-2 Mpro inhibitor.

We thank you, the assay for all compounds including reference compounds have been done in one lab by one hand three times, and not depending upon the reported activity. The reason of such variance might be the strain of the virus used or the condition from other labs.

  1. Lopinavir is an incorrect reference compound as it is a HIV-protease inhibitor and was optimized as such it is not a SARS-COV-2 Mpro inhibitor. Lopinavir in combination with ritonvavir has been used with controversial results to treat COVID 19.

All regards or you, as mentioned before it was a novel scaffold and for the first time was assayed as an antiviral against COVID-19. We used lopinavir here as non-specific protease inhibitor for confirming our mechanism as Mpro protease.

  1. The author should also use the same reference compounds for the biochemical (Table 1) and the cellular assay (Table 2). Remdesivir was used as the reference compound for the cellular assay it would be appropriate to use GC376.

Thanks for you, As known it is important to use specific anti-COVID 19 drug used already in the market to inhibit the progress and multiplication of virus as a reference drug in the study. But the GC376 is a specific main protease inhibitor under development for coronavirus and others.

  1. Also the cellular data does not exhibit the same SAR as the biochemical assay, this should be commented and explained.

Thanks for you, yes somehow it matches the SAR of biochemical assay, and we discussed this in MS as some of the compounds do not work by the same mechanism or affect the viral replication by another way. It was detailed in MS

  1. Generally the work described is rather suited for a short communication than for a full article. The most potent, single digit micromolar SARS-COV-2 Mpro inhibitors disclosed might offer an interesting starting point for further optimization but not more.

Thanks for you, We respect your comment. The study offered different synthetic pathways yielded 17 compounds, different biological assays, molecular modeling approaches, and exhibited good results biological data for further optimization in pharmaceutical chemistry, so it deserves to be published as full article.

Reviewer 2 Report

The research is fairly original and significant and worthy of publication after some extensive revisions, as per my comments below.

I am happy with the characterisation of the compounds.

1. The English is very difficult to follow and extensive editing of language and style is required. I have attached a scan of the first page to indicate the extent of the required corrections. Some extensive proofreading is also required (compound numbers not in bold, see also comments 3 and 6).

2. The keyword needs to Molecular dynamics simulation.

3. Introduction, paragraph 2, last sentence has "nitrnitrogen-containingerocycles".

4. Figure one shows 6 compounds, not 10.

5. Do not use MPro, use main protease (also in the title).

6. Page 2, line 9 from the end has "hepathe titis".

7. The terms fragment-based drug design and lead compound are defined in novel ways in this publication. These definitions are problematic, because these terms are already out there in the literature with different meanings.

8. Page 4: Missing reference for the structure used in docking (and I don't mean just the pdb code).

9. Page 4: Too many detailed results in a section called Aims.

10. Scheme 1: compounds 3a-b need to be identified in step ii.

11. I assume the data in Table 1 are not already published in reference 20. What is meant by saying that they are confirmed as reported in [20]?

12. We need structures of GC-376 and lopinavir. I assume these controls were actually tested here. Or are those literature values? If tested here, some comparison with literature values would be nice.

13. Figure 4 needs a better caption, so it can stand alone. What do the purple, green, red boxes mean etc.

14. Figures 4 and 5 should be swapped. The pyramid belongs under Table 1 as that is also where it is discussed.

15. Page 6, line 148: Should refer to Figure 6.

16. Docking protocol needs to be described in more detail in the methods. Was a control docking performed?

17. The simulations are not well described and too short to be meaningful, unless several replicates were done. The method in the Supplementary says 3 replicates were done, but the text only describes one.

18. Figures 7-10 need much better captions. What exactly is shown? Every symbol needs to be explained. For example, are RMSD values given for all atoms or C alpha atoms?

19. Similarly for Table 3: it needs more footnotes. What are TPSA, ABS, M, T, I, R etc?

Author Response

Comments and Suggestions for Authors

The research is fairly original and significant and worthy of publication after some extensive revisions, as per my comments below.

I am happy with the characterisation of the compounds.

  1. The English is very difficult to follow and extensive editing of language and style is required. I have attached a scan of the first page to indicate the extent of the required corrections. Some extensive proofreading is also required (compound numbers not in bold, see also comments 3 and 6).

Thanks for you, we subjected the MS for extensive editing of language and style and the revised version became ready for publication.

  1. The keyword needs to Molecular dynamics simulation.

Thanks for you, it was added.

  1. Introduction, paragraph 2, last sentence has "nitrnitrogen-containingerocycles".

It was corrected.

  1. Figure one shows 6 compounds, not 10.

It was corrected.

  1. Do not use MPro, use main protease (also in the title).

Thanks for you, we replaced all terms by main protease.

  1. Page 2, line 9 from the end has "hepathe titis".

It was corrected.

  1. The terms fragment-based drug design and lead compound are defined in novel ways in this publication. These definitions are problematic, because these terms are already out there in the literature with different meanings.

All regards to you, We meant here by a fragment-based approach the molecular hybridization protocol of different azoles; pyrazole-linker-triazole with terminal aromatic fragments. The precursor compounds of antiviral activity with different pharmacophoric fragments inspired us to do such study. So, we referred here to fragment and lead terms to such design and might be different somehow from basic terms.

  1. Page 4: Missing reference for the structure used in docking (and I don't mean just the pdb code).

Ok, we added the reference for it in MS. (Douangamath, A., Fearon, D., Gehrtz, P. et al. Crystallographic and electrophilic fragment screening of the SARS-CoV-2 main protease. Nat Commun 11, 5047 (2020). https://doi.org/10.1038/s41467-020-18709-w)

  1. Page 4: Too many detailed results in a section called Aims.

Thanks for you, we summarized this part to be convinced in detail and met the aim of the work.

  1. Scheme 1: compounds 3a-b need to be identified in step ii.

The scheme was redrawn in a clear manner, and 3-ab compounds appeared.

  1. I assume the data in Table 1 are not already published in reference 20. What is meant by saying that they are confirmed as reported in [20]?

Thanks for your accurate inspection, the data here from chemistry to biology are novel and NOT reported before, we mean here by this sentence that “These findings were confirmed the antiviral activity of such Cl-phenylpyrazole scaffold (substituted) better than plan phenylpyrazole itself as reported in our previous study [22]” and this have been published in our last paper ONLY for the phenylpyrazole scaffold.

  1. We need structures of GC-376 and lopinavir. I assume these controls were actually tested here. Or are those literature values? If tested here, some comparison with literature values would be nice.

I thank you very much, the controls were tested in our lab, and discussion about their performance compared to reported work in MS. In addition, the two structures of them were embedded in the MS.

  1. Figure 4 needs a better caption, so it can stand alone. What do the purple, green, red boxes mean etc.

I thank you very much, the boxes were removed from figure 4. The caption was modified in good way.

  1. Figures 4 and 5 should be swapped. The pyramid belongs under Table 1 as that is also where it is discussed.

They were changed to more clear ones and added in the MS.

  1. Page 6, line 148: Should refer to Figure 6.

It was corrected.

  1. Docking protocol needs to be described in more detail in the methods. Was a control docking performed?

I thank you very much, the methodology of docking was rewritten in more detail manner. In addition, the bound ligand within pdb structure was used as control by redocking and rescored to optimize the placement method, the results were discussed in MS and figure 7 showed the comparison.

  1. The simulations are not well described and too short to be meaningful, unless several replicates were done. The method in the Supplementary says 3 replicates were done, but the text only describes one.

All regards for you, the MD simulation are described in shortly in MS and more details are presented in supporting materials and referred for that in main manuscript.  The correct is 3 triplicates NOT one and corrected in MS.

  1. Figures 7-10 need much better captions. What exactly is shown? Every symbol needs to be explained. For example, are RMSD values given for all atoms or C alpha atoms?

Figures captions of 7-10 were rewritten in much manner for clarifying the plots. RMSD values are given for all atoms.

  1. Similarly for Table 3: it needs more footnotes. What are TPSA, ABS, M, T, I, R etc?

Thanks for you, all terms in Table 3 are explained in footnotes.

Reviewer 3 Report

Musa and co-workers reported synthesizing phenylpyrazolone-1,2,3-triazole hybrids via click chemistry using a fragment-based drug design to find potent antiviral agents with promising anti-SARS-CoV-2-MPro inhibition. The fragment-based approach looks interesting and attractive to the readers. This work is suitable for publication in Pharmaceuticals Journal.

Here are some significant comments below.

1.

Abstract: Use bold for compound 6h. Moreover, please check the punctuation mark between the sentence (line # 5) “SARS-CoV-2 virus-infested Vero cells. with different”

2.

So far, several research groups have published nanomolar active compounds, but I have yet to see the potency of the reported compounds. How do the authors explain the novelty of the work?

3.

The authors should write the biological importance of privileged structures of 1,2,3-triazole heterocyclic motifs. Please cite the following article describing how the 1,2,3-triazole play a significant role in drug discovery. EJMC 188 (2020) 111974 (https://doi.org/10.1016/j.ejmech.2019.111974).

4.

Please remove the bold for Figures numbers (Figure 1, 2….n) in the text throughout the manuscript.

5.

Results and discussion (Chemistry): I have not seen the 3a-b precursors in the manuscript, so please recheck it.

6.

Scheme 3 (5h-o): Substituent R2 and N-H is bold; please remove it.

7.

Supporting information (Spectra):  The authors did not include the spectra for all the compounds. I've only seen a few molecules, and 13C spectra have vanished. Figures S5 and S7, which show a 1H spectrum of compound 6n, appear to be the same. Please double-check it.

Author Response

Comments and Suggestions for Authors

Musa and co-workers reported synthesizing phenylpyrazolone-1,2,3-triazole hybrids via click chemistry using a fragment-based drug design to find potent antiviral agents with promising anti-SARS-CoV-2-MPro inhibition. The fragment-based approach looks interesting and attractive to the readers. This work is suitable for publication in Pharmaceuticals Journal.

Here are some significant comments below.

We want to thank the reviewer very much for supporting our manuscript as well as for his constructive and competent criticism. We quoted your entire comments and replied to them step-by-step in blue color under each comment.

1.Abstract: Use bold for compound 6h. Moreover, please check the punctuation mark between the sentence (line # 5) “SARS-CoV-2 virus-infested Vero cells. with different”.

Done

  1. So far, several research groups have published nanomolar active compounds, but I have yet to see the potency of the reported compounds. How do the authors explain the novelty of the work?

 Regarding to our work here, it sounds novel as we introduced a new scaffold design, as antiviral agents targeting protease, with two different heterocyclic motifs; pyrazole and triazoles as promising parts in antiviral drug discovery. Finally, we have two derivatives with promising antiviral activity and close to be optimized in the future to get more potent lead compounds.

  1. The authors should write the biological importance of privileged structures of 1,2,3-triazole heterocyclic motifs. Please cite the following article describing how the 1,2,3-triazole play a significant role in drug discovery. EJMC 188 (2020) 111974 (https://doi.org/10.1016/j.ejmech.2019.111974).

This class of nitrogenous heterocycles plays a substantial role in the development of druggable candidates because of its ability to form hydrogen bonds with certain biological targets. As well, 1,2,3-triazoles resist metabolic degradation in relative to other heterocycles [1,2].

  1. Please remove the bold for Figures numbers (Figure 1, 2….n) in the text throughout the manuscript.

Done all over the manuscript.

  1. Results and discussion (Chemistry): I have not seen the 3a-b precursors in the manuscript, so please recheck it.

Thank you for the careful observation. Numbers 3a & 3b refer to the reactants used in the second step in Scheme 1 (2,5-disubstituted pyrazol-3-one). The proper edit has been made. Kindly refer to Scheme 1 in the updated version.

  1. Scheme 3 (5h-o): Substituent R2 and N-H is bold; please remove it.

Done.

  1. Supporting information (Spectra): The authors did not include the spectra for all the compounds. I've only seen a few molecules, and 13C spectra have vanished. Figures S5 and S7, which show a 1H spectrum of compound 6n, appear to be the same. Please double-check it.

All the spectral data HNMR and CNMR for target compounds were placed in the updated version of supporting materials. The duplicate of 1H spectrum was removed.

Reviewer 4 Report

The authors describe a series of phenylpyrazolone with 1,2,3 triazole and lipophilic aromatic terminals  for dual antiviral and Mpro inhibition. Some of the analogs had greater activity than the reference inhibitors. Most of the compounds exhibited both viral growth inhibition and higher anti-Mpro activity compared to positive control drugs. Compound 6i is indicated to have considerable potential as new pyrazolone-1,2,3-triazole hybrid as the lead compound for the discovery of novel antiviral drugs for COVID-19 infection.

In conclusion, this manuscript contributes to the understanding and development of COVID-19 Mpro inhibitor that will be helpful to the researchers in the field. So, I am supportive of the manuscript for publication in Pharmaceuticals. However, the following are revisions necessary;

1.     In the Scheme 2. Compounds 6a-6f, authors need to label which position of substitution on R2

2.     Please provide the NMR proton data for all the new compounds.

Author Response

The authors describe a series of phenylpyrazolone with 1,2,3 triazole and lipophilic aromatic terminals  for dual antiviral and Mpro inhibition. Some of the analogs had greater activity than the reference inhibitors. Most of the compounds exhibited both viral growth inhibition and higher anti-Mpro activity compared to positive control drugs. Compound 6i is indicated to have considerable potential as new pyrazolone-1,2,3-triazole hybrid as the lead compound for the discovery of novel antiviral drugs for COVID-19 infection.

In conclusion, this manuscript contributes to the understanding and development of COVID-19 Mpro inhibitor that will be helpful to the researchers in the field. So, I am supportive of the manuscript for publication in Pharmaceuticals. However, the following are revisions necessary;

We want to thank the reviewer very much for supporting our manuscript as well as for his constructive and competent criticism. We quoted your entire comments and replied to them step-by-step in blue color under each comment.

  1. In the Scheme 2. Compounds 6a-6f, authors need to label which position of substitution on R2

Done for both substituents.

  1. Please provide the NMR proton data for all the new compounds.

All NMR data are attached in supplementary file.

Reviewer 5 Report

The study is scientifically sound however needs thorough proof reading. Please, address the following issues

Figure 1 needs redrawing; the double bond in the triazole ring looks like two dots rather than a double bond. 

Please report EC50 values to significant number (up to two decimal points)

Figure 3 needs to be redrawn; circles in triazole rings are overlapping bonds.

Hybridisations are represented as sp2 and not as Sp2. 

Figure 4 needs to be redrawn. Remove shadows from boxes. Is there any relevance of using different color boxes; if not, please make them of same color.

Redraw Figure 6; simple rectangular bars look much better.

mL is represented as "mL" and not ml

13C resonances are reported to 1 decimal point.

Author Response

Comments and Suggestions for Authors

The study is scientifically sound however needs thorough proof reading. Please, address the following issues

We want to thank the reviewer very much for supporting our manuscript as well as for his constructive and competent criticism. We quoted your entire comments and replied to them step-by-step in blue color under each comment.

Figure 1 needs redrawing; the double bond in the triazole ring looks like two dots rather than a double bond.

This was due to the limited distances between the two adjacent nitrogens in the triazole ring. However, we removed the bold characters, and hence the double bond could be now easily recognized.

Please report EC50 values to significant number (up to two decimal points)

Done

Figure 3 needs to be redrawn; circles in triazole rings are overlapping bonds.

Done

Hybridisations are represented as sp2 and not as Sp2.

Done

Figure 4 needs to be redrawn. Remove shadows from boxes. Is there any relevance of using different color boxes; if not, please make them of same color.

Redrawn and modified with the same color

Redraw Figure 6; simple rectangular bars look much better.

Replaced

mL is represented as "mL" and not ml

Corrected

13C resonances are reported to 1 decimal point.

It was done in the spectral data.